# Superlinear scaling of riverine biogeochemical function with watershed size

Wilfred M. Wollheim [1✉], Tamara K. Harms [2], Andrew L. Robison [1,7], Lauren E. Koenig [3], Ashley M. Helton[3], Chao Song [4], William B. Bowden[5] & Jacques C. Finlay [6]

River networks regulate carbon and nutrient exchange between continents, atmosphere, and oceans. However, contributions of riverine processing are poorly constrained at continental scales. Scaling relationships of cumulative biogeochemical function with watershed size (allometric scaling) provide an approach for quantifying the contributions of fluvial networks in the Earth system. Here we show that allometric scaling of cumulative riverine function with watershed area ranges from linear to superlinear, with scaling exponents constrained by network shape, hydrological conditions, and biogeochemical process rates. Allometric scaling is superlinear for processes that are largely independent of substrate concentration (e.g., gross primary production) due to superlinear scaling of river network surface area with watershed area. Allometric scaling for typically substrate-limited processes (e.g., denitrification) is linear in river networks with high biogeochemical activity or low river discharge but becomes increasingly superlinear under lower biogeochemical activity or high discharge, conditions that are widely prevalent in river networks. The frequent occurrence of superlinear scaling indicates that biogeochemical activity in large rivers contributes disproportionately to the function of river networks in the Earth system.

[1] Department of Natural Resources and the Environment, University of New Hampshire, Durham, NH 03824, USA. [2] Department of Biology and Wildlife and Institute of Arctic Biology, University of Alaska Fairbanks, Fairbanks, AK 99775, USA. [3] Department of Natural Resources and the Environment, and the Center for Environmental Sciences and Engineering, University of Connecticut, Storrs, CT 06269, USA. [4] Department of Earth and Environmental Sciences, Michigan State University, East Lansing, MI 48824, USA. [5] Rubenstein School of Environment and Natural Resources, University of Vermont, Burlington, VT 05405, USA. [6] Department of Ecology, Evolution and Behavior, University of Minnesota, St. Paul, MN 55108, USA. [7] Present address: Stream Biofilm and Ecosystem Research Laboratory, École Polytechnique Fédérale de Lausanne, Lausanne, Switzerland. ✉email: wil.wollheim@unh.edu

River networks regulate biogeochemical fluxes from continents to the oceans[1–3] and to the atmosphere[4–9], influencing water quality, coastal dead zones, food webs, and greenhouse gas emissions. For example, at a continental scale, inland waters return ~25% of net carbon uptake by terrestrial ecosystems back to the atmosphere[10]. Similarly, denitrification in river networks removes 20–50% of nitrogen inputs, reducing transfer to oceans while also accounting for a significant proportion of global nitrous oxide emissions[5,11]. However, reducing uncertainty in estimates of cumulative biogeochemical function of whole river networks remains a challenge due to limited sets of nested observations within river networks that capture variation in channel size and stream flow, which span orders of magnitude[12–14].

Allometric scaling with respect to watershed size offers a synthetic approach to estimating cumulative biogeochemical function by integrating key characteristics of river networks, including network structure (i.e., shape), channel hydraulics (i.e., widths), and biogeochemical activity[15]. This approach is analogous to allometric scaling applied to explain metabolic function with increasing size from individual organisms to whole ecosystems[16–18]. In each case, attributes of the distribution network result in transport limitation that causes a scaling pattern spanning orders of magnitude in body or ecosystem size. For example, the Metabolic Theory of Ecology demonstrates how metabolism in individual organisms increases predictably as a power function of body size, with sublinear scaling (log-log slope <1) due to transport constraints through fractal circulatory networks[19]. Sublinear allometric scaling has also been observed for individual lakes and estuaries, where cumulative metabolism increases more slowly than size of the water body due to transport limitation of energy or nutrients[17,18,20]. River networks are also fractal transport networks[21,22] and might exhibit similar allometric scaling relationships, though key differences from circulatory systems of organisms or estuaries suggest that scaling of cumulative biogeochemical function of entire river networks might deviate from the sublinear scaling observed in these systems.

The fractal nature of river network structure leads to predictable scaling of many physical characteristics of river networks. Scaling of attributes such as channel length and width are well known[21,22] and have implications for biogeochemical function of entire river networks[15,23,24]. For example, power law scaling describes the relationship of watershed area with cumulative gross primary production and proportional removal of dissolved organic carbon due to interactions among biological activity, network structure, and river hydraulics[15,23,24]. However, it remains unclear why these patterns arise, and importantly how they are influenced by temporal variation in river discharge, a key influence on riverine function[13,25] that is highly influenced by climate change, river regulation, and water use[26,27]. Here, we use a river network modeling approach to quantify cumulative biogeochemical function of river networks as their watershed size increases. We demonstrate how mechanistic interactions among hydrological conditions, network structure, and rates of biogeochemical processes cause variation in allometric scaling relationships, and the implications of allometric scaling for the role of river networks in the Earth system.

**Conceptual model for allometric scaling of cumulative riverine function.** For any given biogeochemical process, cumulative riverine function ($F$, mass time$^{-1}$) throughout an entire river network increases with its watershed area ($A$, km$^2$). The increase within an individual river network from the average headwater catchment to the entire watershed defined by a basin mouth can be described with a power relationship:

$$F = cA^d \qquad (1)$$

where, $d$ is the rate of increase in cumulative riverine function with watershed area (i.e., allometric scaling) and $c$ is the normalization constant, equivalent to the cumulative riverine function in a headwater river network with $A = 1\,km^2$ (Fig. 1). The scaling of cumulative function (Fig. 1C) accounts for changes in local areal process rates per unit stream surface area with increasing stream size (local scaling = $m.local$, Fig. 1A), and accumulation of surface area ($SA$) as watershed area increases (Fig. 1B). Surface area can be either the stream bottom (benthic) or at the air-water interface depending on the biogeochemical process being considered. Parameters describing these components are well constrained, which we implement in an existing river network model to quantify allometric scaling in river networks with a range of potential characteristics (e.g., shape, hydraulics, discharge, biogeochemical activity; see Methods). In this framework, we consider processes that are either not relevant (e.g., gas exchange, particle settling) or significant in the water column volume[28].

We demonstrate allometric scaling assuming spatially homogenous inputs of water (i.e., runoff; depth time$^{-1}$) and non-point sources (mass area$^{-1}$ time$^{-1}$), while local aquatic process rates may vary along gradients according to river size (Fig. 1A). Runoff then accumulates to determine discharge (volume time$^{-1}$) based on drainage area for rivers of increasing size. While the assumption of uniform runoff is reasonable[29,30], uniform non-point sources is likely often violated in real watersheds[31]. However, this assumption allows us to assess effects of network structure and hydrological conditions on cumulative biogeochemical function (Eq. 1). The rate at which non-point material fluxes in the river network are modified by riverine processes depends on the type of constituent (gas, particle, dissolved) and their associated process rates. We examined heterogeneity in biogeochemical process rates along gradients in river size[32], considering scenarios of increasing, decreasing, or constant process rates with size (defined by local process scaling, $m.local$). The analysis thus provides the underlying scaling tendency given a river network structure, which can serve as a useful null model against which to compare patterns of heterogeneity in loading or instream process rates caused by ambient or human factors in particular river networks.

## Results and discussion

**Superlinear scaling of cumulative river network function with watershed area.** Geomorphological characteristics of river networks (i.e., their structure) determine the distribution of river lengths, which establish the terrestrial-aquatic interface and therefore constrain the distribution of initial material loading from the landscape to rivers of different size. Typically, total inputs to the river network are distributed according to the proportion of length in each river order[22,33]. River length along with river hydraulics (river width) define the distribution of benthic (stream bottom) or air-water surface area (SA), where most biogeochemical processes occur in rivers[28]. Thus, length and SA scaling underpin allometric scaling of cumulative biogeochemical function in river networks (Fig. 1).

In a diverse set of river networks distributed across multiple biomes (Supplementary Table 2), cumulative river length scaled nearly linearly with increasing watershed area (mean length slope = 1.06, range 0.99–1.16) (Supplementary Fig. 1A) implying linear scaling of material inputs to river networks from surrounding land. In contrast, cumulative SA in observed networks scaled superlinearly (mean SA slope = 1.23, range 1.1–1.4) (Supplementary Fig. 1B). Theoretical networks of different shapes representing common structures[23] (Supplementary Tables 1 and 3) generated physical scaling relationships that

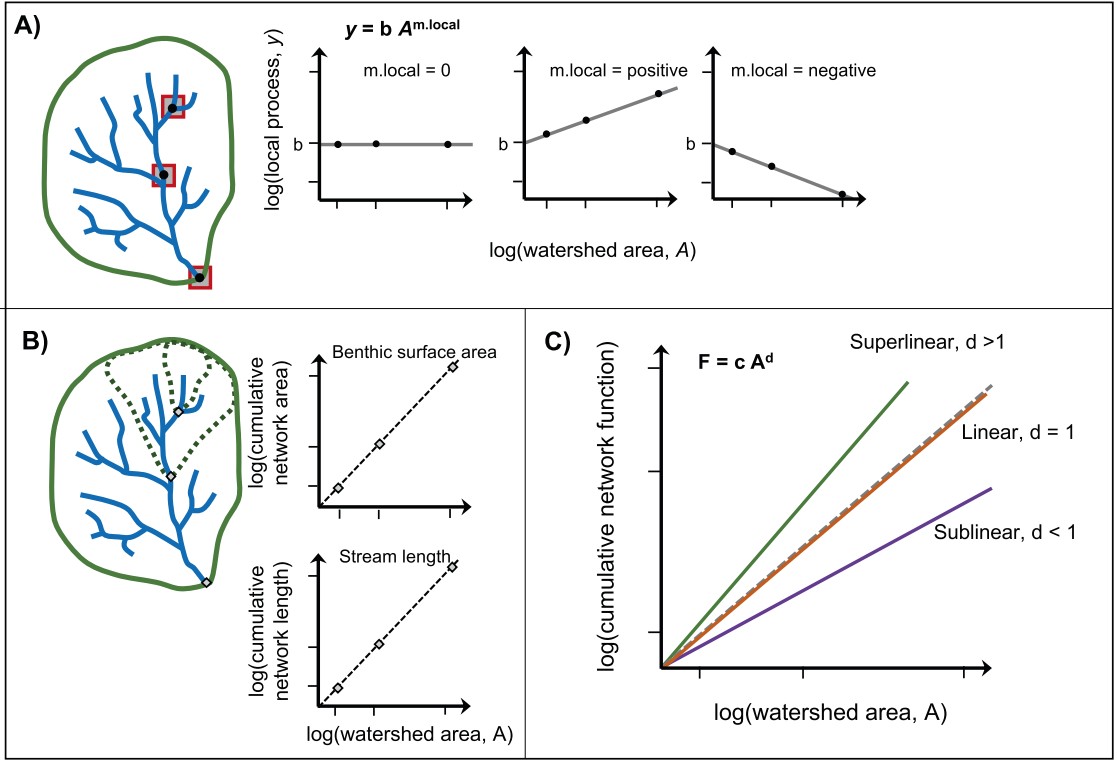

**Fig. 1 Conceptual model of allometric scaling.** Allometric scaling ($d$) of cumulative biogeochemical function ($F$) in river networks with increasing watershed area as determined by local areal process rates ($y$) and physical structure of river networks. **A** Scenarios describing the change in local process rate (as either mass surface area$^{-1}$ time$^{-1}$ or as length time$^{-1}$) in stream reaches (local areal process slope = $m.local$, $b$ = constant, Eq. (2)) vs. watershed area draining to the stream location. **B** Scaling of cumulative benthic surface area vs. watershed area and stream length vs. watershed area, accounting for the entire upstream river network. Scaling of cumulative length and surface area can differ. **C** Examples of scaling of cumulative biogeochemical function over the entire river network (mass time$^{-1}$) with increasing watershed area (cumulative function slope = $d$, Eq. (1)) resulting from variation in both local process rates (**A**) and cumulative physical attributes of networks (**B**).

were similar to those observed (See Supplementary Note 1, Supplementary Fig. 1). Superlinear scaling of cumulative SA occurs because as watershed area increases, larger, wider rivers contribute increasingly to cumulative SA, but relatively less to cumulative length. In the analyses that follow, we explore biogeochemical scaling relationships using the theoretical networks to demonstrate the constraints on allometric scaling of cumulative biogeochemical function, given a range of empirically-supported hydrologic and biogeochemical scenarios (Supplementary Tables 1 and 5).

Allometric scaling relationships between cumulative biogeochemical function and watershed size ranged from linear to superlinear, depending on biogeochemical activity and hydrological conditions. The simplest biogeochemical scenario describes an areal process rate that is independent of substrate concentration (i.e., zero-order kinetics), such as is commonly used to represent aquatic primary production (GPP) or ecosystem respiration (ER)[34–36]. When zero-order reaction rates remain constant with increasing river size ($m.local = 0$, Fig. 1A), allometric scaling of cumulative riverine function is simply identical to that of the scaling of SA as watershed size increases (i.e., superlinearly; cumulative function scaling slope, $d = 1.23$, range of 1.11–1.41, Fig. 2). However, areal processes such as GPP and ER typically increase with watershed size (i.e., $m.local > 0$, Finlay 2011, Supplementary Fig. 2), indicating higher rates in larger rivers. If $m.local = 0.5$, cumulative function scales highly superlinearly (Fig. 2; mean $d = 1.64$, with range 1.47–1.87). If the local process rate declines with increasing watershed area ($m.local = -0.5$), cumulative function scales linearly with watershed size ($d = 1.0$, range of 0.95–1.09). Thus, a tendency for superlinear

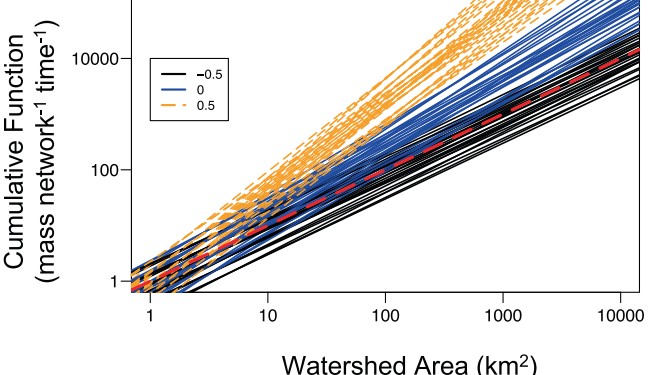

**Fig. 2 Allometric scaling for zero-order processes.** Cumulative biogeochemical function vs. watershed area when aquatic processes are zero-order, wherein material concentrations in the water column do not influence local areal process rates. Each line represents a different stream network, characterized by distinct network geomorphology and river hydraulics ($n = 27$) for each value of local scaling ($m.local$); network parameters used to generate cumulative functions are described in Supplementary Table 1. Line colors represent potential local scaling of areal process rate with increasing river size ($y = bA^{m.local}$), where $m.local = -0.5$ (black), 0 (blue), or 0.5 (orange dash), representing declining, constant, or increasing local rates, respectively, with increasing river size. Red dashed line indicates linear cumulative scaling. Superlinearity occurs due to scaling of benthic surface area (Supplementary Fig. 1) and is attenuated or accentuated by the relationship of local process rate with river size.

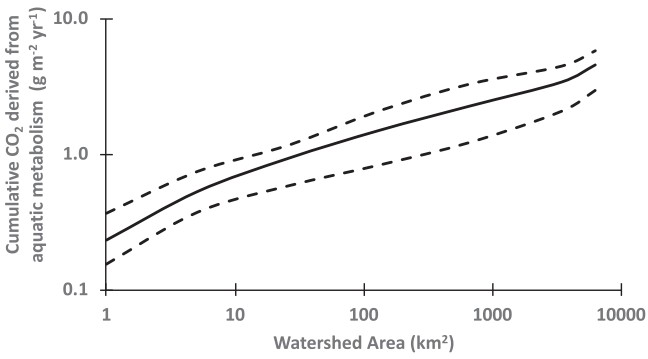

**Fig. 3 Allometric scaling of net carbon production with increasing watershed area.** Cumulative $CO_2$ derived from aquatic metabolism (cumulative ER – cumulative GPP, normalized to watershed area). Model results incorporate observed trends in the local rate of GPP and ER with watershed area (Supplementary Fig. 2) for a rectangular river network at mean annual flow (500 mm $yr^{-1}$). Median (solid line), 25th percentile and 75th percentiles (dashed lines) are derived from 9 model scenarios that reflect potential variation in hydraulic dimensions (varying $e$ and $f$ in Supplementary Table 1).

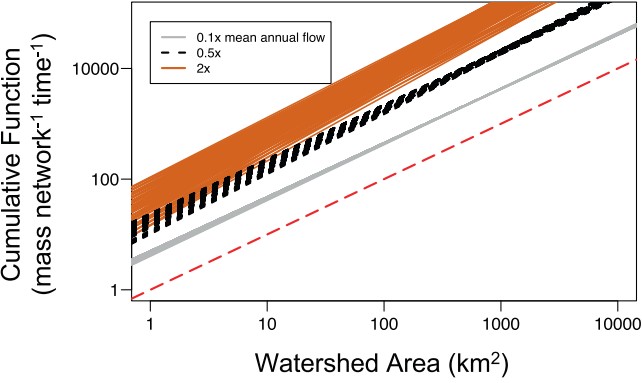

**Fig. 4 Allometric scaling for first-order processes.** Cumulative biogeochemical function vs. watershed area when aquatic processes (as $v_f$) are first-order, wherein local areal process rates are influenced by concentration of substrate in the water column. Colors represent flow conditions as a proportion of mean annual flow: 0.1x mean annual flow (gray), 0.5x mean annual flow (black dashed), and 2x mean annual flow (orange). Red dashed line indicates linear scaling. Each line represents a distinct scenario combining variation in river hydraulics and scaling relationships of local areal process rate with increasing river size (*m.local*) for a rectangular network ($n = 81$ for each flow category; parameters described in Supplementary Table 1). Model results reflect average process rate ($v_f$) in a 1 $km^2$ headwater watershed = 500 m $yr^{-1}$ ($b$ in Eq. (2)). Scaling of cumulative function becomes increasingly superlinear and variable at higher flows.

scaling of cumulative function arises from the scaling of SA (superlinear) that can be dampened or enhanced depending on how local process rates change with river size (defined by *m.local*).

Superlinear scaling of cumulative biogeochemical function indicates that larger watersheds contribute proportionally more per unit land area than smaller watersheds to the continental-scale function of rivers within the Earth system. To illustrate, we applied the scaling model with empirically derived local scaling terms for annual mean areal GPP (*m.local* = 0.49) and ER (*m.local* = 0.22) synthesized for rivers with watershed area up to 10,000 $km^2$ (Supplementary Fig. 2, data from[35]). The resulting allometric scaling parameters predicted that cumulative riverine $CO_2$ production (=ER − GPP) from a rectangular river network (Supplementary Table 3) would increase from a median of 0.23 g $m^{-2}$ $yr^{-1}$ (per unit watershed area) in a headwater catchment (drainage area [A] = 1 $km^2$) to 4.6 g $m^{-2}$ $yr^{-1}$ in a 7th-order river network (A = 6321 $km^2$) (Fig. 3). This allometric estimate of $CO_2$ generated by riverine metabolism for a 7th-order watershed is comparable to a global estimate of 6.7 g $m^{-2}$ $yr^{-1}$ (per unit watershed area) derived from estimates of total riverine emissions of $CO_2$ (24 g $m^{-2}$ $yr^{-1}$)[37] and a global estimate of riverine contribution to $CO_2$ flux (28%)[9]. The allometric scaling model demonstrates that although local rates of GPP increase proportionally more than ER with increasing watershed area (local scaling for GPP > local scaling for ER, Supplementary Fig. 2), the absolute amount of cumulative net riverine $CO_2$ production increase faster than watershed size because larger rivers continue to be net heterotrophic and SA scales superlinearly with watershed area. We conclude that because of combined effects of superlinear scaling of SA and a tendency for increasing GPP and ER with stream size (*m.local* > 0), the contribution of riverine processes to watershed-scale $CO_2$ emissions scales superlinearly with watershed area. Therefore, $CO_2$ generated by river networks of larger watersheds contributes relatively more to global $CO_2$ emissions than $CO_2$ generated by smaller watersheds on a per unit land basis.

**Substrate limitation influence on allometric scaling in river networks**. The biogeochemical scenario characterized by zero-order kinetics does not account for the possibility of dynamic substrate limitation that underlies many scaling relationships[18,20]. For many biogeochemical processes, including denitrification, nutrient assimilation, or gas exchange, areal process rate changes

with substrate concentration (i.e., first-order kinetics). When processes are concentration-dependent, allometric scaling of cumulative network function is influenced by downstream substrate limitation, which in turn depends on hydrologic conditions (discharge) and the intensity of biogeochemical activity.

Allometric scaling of cumulative function for substrate-limited processes shifts from linear under low flows to superlinear under high flows (Figs. 4 and 5, Supplementary Fig. 5). We demonstrate this dynamic using a scenario parameterized to the assimilation of nitrate, a first-order process that is constant with river size (*m.local* = 0)[38,39] and typically has an observed rate intermediate to other measured in-stream processes (uptake velocity, $v_f$ = ~500 m $yr^{-1}$ Supplementary Fig. 4). The uptake velocity ($v_f$) parameter allows explicit examination of the effects of biogeochemical process rates because it is mathematically independent of discharge, unlike time constants (k, $time^{-1}$)[40,41] (see Methods). When flow and associated material supply from the landscape are low (0.1× mean annual flow), scaling of cumulative function, $d$, approaches a lower limit set by the scaling of stream length with watershed size ($d$ ~ 1) (Fig. 5b and Supplementary Fig. 5). That is, material inputs from land are removed immediately upon entering the river network. At low flows, varying assumptions of hydraulic geometry (i.e., rate of width change) introduces little variation in the scaling exponent (mean $d$ = 0.98, range 0.98–1.0, Fig. 4, Supplementary Fig. 5). As flows increase, scaling of cumulative function is increasingly superlinear, and at very high flow (20x mean annual flow), $d$ approaches the limit set by SA scaling, which is defined by river network structure and river hydraulics (mean $d$ = 1.16). Further, the scaling parameter ($d$) was most variable at high flows across scenarios of river hydraulics (Eq. (3)) for a given network structure (range from 1.10 to 1.24 for rectangular network, Fig. 4; Supplementary Fig. 5), indicating physical characteristics are more important determinants of cumulative biogeochemical function under high-flow conditions.

The shift from linear to superlinear scaling with increasing flow is explained by the changing spatial distribution within the

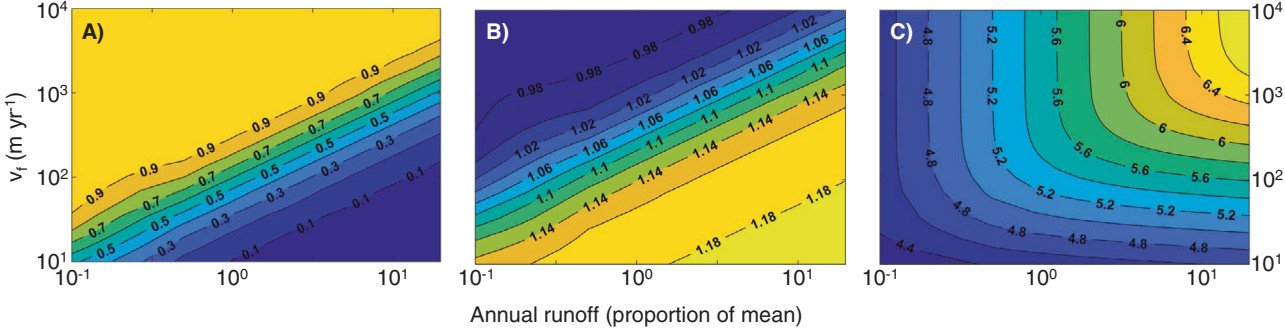

**Fig. 5 Variation in removal proportion, allometric scaling, and cumulative function.** Model scenarios describing effects of local process rate (as uptake velocity, $v_f$) and discharge (as a proportion of mean annual discharge) on (**A**) mean network-scale removal (proportion of inputs from land removed by the entire river network), (**B**) mean cumulative scaling slope, $d$, and (**C**) cumulative aquatic function over the entire river network ($\log_{10}$ kg of material yr$^{-1}$ in the modeled river network). Each combination of $v_f$ and discharge is based on the mean of 27 hydraulic scenarios for a rectangular network as described in Supplementary Table 1. Contour intervals indicate the value for each isobar boundary.

network of riverine uptake relative to biogeochemical inputs from the landscape. Under the assumption of homogeneous, non-point source inputs, material inputs to river networks occur predominantly in headwater streams because of greater total channel lengths compared to large rivers, thereby intersecting more of the landscape and intercepting terrestrial runoff[33] (Supplementary Table 3). Under low flow, material inputs to the river network are low relative to riverine demand, so materials are removed near their point of entry (headwater streams) and little is transported downstream[13,25], resulting in linear scaling of cumulative function with increasing watershed area. In such a scenario, overall removal proportion by the river network is nearly complete (Fig. 5A) and processing in larger rivers is limited by low material inputs from upstream. The scaling relationship at low flow varies little across potential scenarios of river hydraulics and network structure because materials are removed immediately upon input to the network, regardless of network shape or hydraulic parameters.

In contrast, at high flow, material inputs overwhelm local demand or uptake in headwater streams due to increased water velocity, short flow path length, and reduced residence time, resulting in greater material transported from point of input to larger rivers[13]. Scaling of cumulative function for first-order processes then approaches the scaling of SA (Fig. 5b, Supplementary Fig. 5), with each unit area of the river network functioning at its maximum areal processing rate ($U$) as set by the concentration ($C$) of material inputs from the landscape and the uptake velocity ($U = v_fC$). The scaling relationship is more variable across networks of different configuration at high compared to low flow because river network shape and river hydraulics have a relatively large effect on how SA accumulates with watershed size (Supplementary Fig. 1b).

Allometric scaling also shifts from linear to superlinear as the rate of first-order processes (as $v_f$) decreases (assuming $m.local = 0$). A high process rate (i.e., high demand for a material) results in linear scaling of cumulative function because inputs are removed immediately upon entry to the river network, similar to the effect of low flow conditions (Fig. 5, Supplementary Note 2 and Supplementary Fig. 6). Conversely, scaling is increasingly superlinear as the first-order process rate declines, indicating that larger rivers contribute more to river network function when demand for a material is on average lower (Fig. 5 and Supplementary Fig. 6). The effect of process rate on the scaling parameter is similar to that of discharge because the relative rates of processing and transport, as summarized by the Damköhler number[42], define the proportion of the material load processed in a given stream reach or transported further downstream (see

Methods, Eq. (5), where $v_f/H_L$ is the Damköhler number). Process rates (as $v_f$) vary over orders of magnitude depending on the process (Supplementary Fig. 4), indicating that allometric scaling will differ substantially among constituents. Further, if $m.local > 0$ for first-order processes, the tendency towards superlinearity is even greater, as with the zero-order metabolism example described above. For many processes, rates as $v_f$ appear constant with river size[38,39,41,43–45] (i.e., $m.local = 0$). However, local process rates could vary with river size[32] and synthesis of empirical $v_f$ measurements suggest a weak tendency toward increased local process rates with increasing discharge ($m.local > 0$; $r^2 < 0.24$; Supplementary Fig. 4, Supplementary Table 5) suggesting the potential for even greater superlinearity (Supplementary Note 3).

In summary, the process rate determines the range of flows over which superlinear scaling occurs, with lower first-order process rates resulting in a broader range of flows that exhibit superlinear scaling (Fig. 5b, Supplementary Fig. 6). Thus, larger rivers are more important for slower processes (i.e., those with lower $v_f$) across a wider range of flows (e.g., denitrification[3], Supplementary Fig. 4). Note that if scaling of biogeochemical function is in terms of cumulative river network SA rather than watershed size as emphasized here, sublinear scaling occurs under low flow conditions ($d \sim 0.85$) due to transport limitation of processes in large rivers, a result similar to that previously reported for individual lakes and estuaries[18] (see Supplementary Note 4).

**Implications of allometric scaling for riverine contributions to the Earth system.** Allometric scaling relationships provide useful and succinct estimates of material exchange between river networks and the atmosphere (i.e., cumulative flux of gaseous products) and regulation of material fluxes from land to oceans (i.e., mass of material inputs removed from downstream flux). Importantly, model scenarios demonstrated that scaling of cumulative function is dependent primarily on flow and process rate (Fig. 5) rather than network structure (Supplementary Fig. 5), which has significant implications due to temporal variability in discharge and processing rates that span several orders of magnitude (Supplementary Fig. 4). For example, the scaling model indicates that evasion of terrestrially-derived dissolved gases, including $CO_2$ (median $v_f \sim 1500$ m yr$^{-1}$, Supplementary Table 5)[46,47], is nearly complete (>90%) across most flow conditions, and occurs primarily in smaller streams because $d$ remains nearly linear due to gas exchange rate being relatively high (Fig. 5). In contrast, regulation of nitrate fluxes to the ocean through permanent removal by denitrification (median $v_f \sim 25$ m yr$^{-1}$,

Supplementary Table 5) is more dependent on flow conditions (Fig. 5). At very low flows, cumulative removal proportions via denitrification are high (~90%), but $d$ is superlinear across all flow conditions (Fig. 5A, B), indicating that larger rivers play an important role in permanent nitrate removal. We predict similar patterns for fluxes of nitrous oxide resulting from aquatic denitrification[5,48], respiration of refractory DOC, and other constituents with relatively low processing rates.

The allometric scaling model explains why watershed size is often a poor predictor in empirical models of nutrient exports from watersheds to oceans[49,50], despite the importance of in-stream processing[51], the effect of which should become more evident in larger watersheds. Rather than implying that river networks are unimportant in regulating exports from catchments to coasts, our findings suggest watershed size will have little explanatory power in models of downstream material fluxes when local processing rates are very high, or at low flow. Rapid removal near where materials enter the river network will result in no apparent effect of watershed size on export (i.e., $d \sim 1$; Fig. 5). High removal proportions and high scaling exponent do not coincide and would only be detectable over a relatively narrow range of flow conditions for any given process rate (Fig. 5). For example, for an uptake velocity = 100 m yr$^{-1}$, removal at base-flows may be >90% while allometric scaling slope <1.04 so superlinear scaling would not be detectable. At high flows, scaling may be superlinear, but removal proportions relatively low. Only near mean annual flow might removal be high enough to detect superlinear scaling. Because effects of watershed size are usually explored using spatial surveys that require assumptions of stable baseflows[52,53], or by comparisons of mean annual fluxes across watersheds that integrate a wide variety of conditions, the effect of watershed size is rarely identified empirically, despite potentially high cumulative network removal of material inputs. Overall, the superlinear scaling predicted under many model scenarios implies disproportionate contributions of larger rivers to continental-scale material cycling, but with smaller streams contributing disproportionately during times of low discharge or high reaction rates.

**Merging observations with scaling theory.** The range of scaling exponents identified based on scenarios of river network structure, river hydraulics, discharge (runoff) conditions, and biogeochemical activity suggests there is no universal scaling law of cumulative river network function, in contrast to scaling functions applied to individual organisms or ecosystems (Supplementary Notes 4 and 5[16,18]). Further, it is likely we have underestimated the range of scaling exponents, given that the simple scaling model presented omits key factors that influence river network biogeochemistry, including biogeochemical activity in the water column of very large rivers[28,48]; hydrologic dynamics within the streambed[54]; ponded waters (including reservoirs and floodplains)[55,56]; saturating reaction kinetics[3]; stoichiometry[57]; climate regime[58]; and spatially heterogeneous material loading (e.g., greater inputs nearer river mouths, especially in many urbanized watersheds)[31]. Spatial heterogeneity in process rates were addressed in terms of gradients but not patchiness. Previous river network models suggest patchiness that creates local hot and cold spots in terms of reaction rate results in higher network removal, but that the effect is relatively small compared to homogenous biological activity[59]. The modeling framework proposed here could be applied to evaluate these factors using fully spatially-distributed approaches and empirical observations.

Scaling of riverine biogeochemical processes is critical to understanding the role of river networks in the Earth system. Under the assumptions of hydraulic geometry and process rates presented here, individual river networks occupying a continental extent contribute disproportionately to global riverine biogeochemical function on a per unit watershed area basis compared to smaller watersheds that drain directly to the coast, particularly at higher flows. Globally, the largest 101 watersheds drain 65% of the Earth's land mass[12], yet biogeochemical processes in larger rivers remain understudied compared to small streams[39]. The allometric modeling approach provides predictions that can guide the design of monitoring programs needed to quantify cumulative network function. Specifically, enhanced monitoring networks are needed that feature nested observations spanning whole individual networks, high-frequency measurements that account for the full range of flows, and locations capturing the diversity of material inputs (e.g.,[60]). The scaling model presented here predicts that such observations will frequently yield superlinear scaling of cumulative biogeochemical processing due to underlying river network structure, disproportionate material inputs from the landscape in headwaters, upstream-downstream connectivity, and high processing potential downstream, underscoring the role of river networks in functioning of the Earth system.

## Methods

**Overview.** The scaling of cumulative function by an entire river network with increasing watershed area (Eq. (1)) requires scaling both the abiotic template and local areal process rates (Fig. 1). Longitudinal variation of a local areal process rate, $y$ (mass surface area$^{-1}$ time$^{-1}$), as stream size increases with watershed area ($A$) can be modeled with a power function as:

$$y = bA^{m.local} \tag{2}$$

where the constant $b$ represents the local process rate in a stream reach with $A = 1$ km$^2$, and $m.local$ is the scaling exponent that describes how the local areal rate changes with increasing stream size as indicated by $A$ (Fig. 1A). The local process rate, y, can be modeled assuming either zero-order or first-order reaction rates (see below).

Scaling of cumulative biogeochemical function in stream networks also depends on the scaling of cumulative benthic or water surface area (SA) (Fig. 1B), because aquatic processes occur primarily across these interfaces of all but the largest rivers[28,40,41]. The fractal nature of river networks results in cumulative channel length and SA increasing as a power function of watershed area (Fig. 1B) as influenced by geomorphic attributes of the network structure (e.g., shape, number and length of streams in each river order)[23,24]. Scaling of cumulative length is due to structural characteristics of the network alone, whereas SA is also influenced by changes in wetted width as a function of mean discharge that accumulates with watershed area. The relationship between mean annual channel width ($w$) and mean annual discharge ($Q$) in a downstream direction is described by:

$$w = eQ^f \tag{3}$$

where $f$ typically varies between 0.4 and 0.6[61,62] and $e$ is the normalization constant. Temporal variation in discharge has a relatively small influence on channel width because the change with discharge at-a-site (in contrast to discharge in the downstream direction) typically has an exponent of only 0.1, until bank full-flood stage is reached[62]. River network structure also determines flow path probabilities[22], which in turn influence biogeochemical function scaling with increasing watershed size.

We quantified the scaling of cumulative biogeochemical function over entire river networks with increasing watershed area using a previously developed statistical river network model[13,25,41]. The model was applied using a range of model parameters constrained by previous studies (Supplementary Table 1) to demonstrate the potential envelope of allometric scaling relationships. The model accounts for river network characteristics, including river network structure, the probability of flow paths (upstream-downstream connectivity), flow variability, river hydraulics, and aquatic biogeochemical processes (see details below).

We used 2 types of scenarios to represent biogeochemical process behavior throughout river networks to demonstrate allometric scaling outcomes. The first approach used an areal process rate that is not influenced by local reactant concentration and therefore independent of upstream inputs (i.e., zero-order process rates) but potentially allowed to vary with stream size (Eq. (2)). The second approach applied first-order reaction kinetics as is commonly assumed in network models of sediment, gas, nutrient, or carbon flux[40,56]. In this approach, a process rate as uptake velocity ($v_f$, L T$^{-1}$) is applied, where the local areal process rate is determined by concentration of reactant in the water column (see details below). As a result, the local areal uptake within any given stream reach is influenced by the impact of all upstream processes on downstream concentrations. The process rate may also vary with stream size (Eq. (2))[44]. We applied heuristic scenarios of

biogeochemical process rates to demonstrate factors controlling scaling relationships, as well as scenarios of empirically derived biogeochemical process rates that explore the implications of the scaling approach for understanding network-scale function. To demonstrate the underlying cumulative scaling properties of river networks, the model scenarios also assumed that terrestrial runoff and associated constituent concentrations are spatially uniform.

**Physical structure.** River network structure is defined by the number, mean length, and mean watershed areas of streams based on their stream order (when two branches of the same order converge, a higher order is created)[22,63]. These characteristics are estimated for each stream order in the model using geomorphological parameters: number ratio $R_b$; length ratio $R_l$; area ratio $R_a$; watershed area and mean length of a first-order stream $A_1$ and $L_1$, respectively (e.g., Supplementary Table 3). The distributions of where water and non-point material inputs from land initially enter each stream order within any given watershed are estimated from the proportion of total watershed area draining 1st order catchments and for higher-order streams, the proportion of total stream length in each higher-order stream (e.g., Supplementary Table 3). Streams of a given order flow into higher-order streams according to probabilities using the Geomorphic Unit Hydrograph approach[22] which define flow paths through the river network (e.g., Supplementary Table 4). Flow path probabilities are critical in scaling of riverine biogeochemical function because they determine how surface area and biogeochemical processes accumulate with increasing watershed area. They also influence how upstream processes influence the downstream availability of reactants for biogeochemical processes that are concentration dependent.

To explore how scaling potentially varies among different river network structures, we parameterized the statistical model based on the network attributes of three differently-shaped theoretical channel networks[64]. Theoretical channel networks exhibit the fractal properties observed in natural river networks[22] and were used recently to model river network biogeochemical processes[15,23]. We used them here to demonstrate potential variability in allometric scaling due to network structure. We compared the resulting scaling of length and surface area with increasing watershed area from our statistical model as parameterized from the theoretical networks to the length and surface area scaling in observed river networks derived from digital elevation models of previously studied watersheds[65–67] to ensure consistency (Supplementary Note 1, Supplementary Fig. 1). We applied geomorphic parameters from all networks to a consistent watershed area of ~6300 km$^2$ (stream order 6 or 7 at mouth depending on network structure parameters), a watershed area large enough to have a sufficient range of scale to develop the power relationship between cumulative biogeochemical function and watershed area.

In all scenarios, we assumed mean annual runoff over the entire watershed was 500 mm yr$^{-1}$, a reasonable value for temperate watersheds[68]. Mean annual runoff was used to calculate mean annual discharge ($Q$ m$^3$ s$^{-1}$) in each river order using the river order's mean watershed area. Many temperate watersheds exhibit linear scaling of discharge with watershed size[29], consistent with this approach. Mean discharge was used to estimate the mean width ($w$), depth ($h$), and velocity ($v$) of each river order using hydraulic geometry equations that describe how $w$ changes with mean $Q$ in the downstream direction (Eq. (3), Supplementary Table 1)[61,62]. We explored the role of runoff/discharge variability over time for scenarios with first-order kinetics by varying runoff between 0.1x (roughly baseflow) to 20x (extreme flow) relative to the mean annual runoff. Changes in width, depth, and velocity due to variable discharge at local stream reaches (i.e., "at-a-site") were estimated using at-a-site width vs. discharge equations ($g$ in Supplementary Table 1). The combination of river network structure, flow path probabilities, runoff, and river hydraulics determines how reactive surface area scales with increasing watershed area.

**Biogeochemical activity.** We used two approaches to represent biogeochemical activity per unit surface area assuming 1) zero-order processes and 2) first-order kinetic processes. An example of a zero-order process is metabolism (gross primary production and ecosystem respiration), which is often represented in streams and rivers as an areal rate ($U$, M L$^{-2}$ T$^{-1}$) independent of nutrient concentration[35,36]. An example of a first-order process is nitrogen assimilation or denitrification, which is often represented in streams and rivers as a benthic dominated process using the uptake velocity parameter ($v_f$, L T$^{-1}$), which is equivalent to stream bottom $U$ (M L$^{-2}$ T$^{-1}$)/water column concentration ($C$, M L$^{-3}$). Both types of activity are considered local rates (Fig. 1B) that may change with stream size (local scaling) as described in Eq. (2). Cumulative riverine processes were aggregated over all streams within each river order as the product of surface area and areal uptake rate. For the first-order kinetics approach, $U = v_f C$, where $C$ entering each river order is first predicted by the river network model to account for upstream removal. We use the uptake velocity rather than a time-specific process rate ($k$, T$^{-1}$) because $v_f$ is independent of discharge unlike $k$ (which is dependent on water depth for processes at interfaces). $v_f$ is therefore more appropriate for processes occurring across interfaces (water-benthic, water-air) and is reasonably represented as constant across streams of different size[38,39,41,43–45]. The $v_f$ parameter can also be applied to processes affecting different types of constituents (nutrient uptake, sediment settling, gas exchange) so models that incorporate it can be used for comparing the role of river

networks for different constituents. The role of temperature can also be inferred by varying $v_f$ using temperature dependant functions (e.g., Q10).

**River network model.** For first-order process scenarios, upstream processes influence downstream reactant supply and local areal rates. In the model, the average flux ($F$, M T$^{-1}$) exported from rivers of a given river order (which is then routed downstream based on flow path probabilities) is determined as in ref. [25,41]:

$$F = Upstream_{in} * (1 - R_{full}) + (Local_{in} + Tributary_{in}) * (1 - R_{half}) \qquad (4)$$

$$R_x = 1 - exp(-v_f/H_{L\_x}) \qquad (5)$$

$$H_{L\_full} = Q_{mid}/(w_{mid} * L) \qquad (6)$$

$$H_{L\_half} = Q_{down}/(w_{down} * (L/2)) \qquad (7)$$

where $Upstream_{in}$ is the upstream input from the two rivers that initially create the order (M T$^{-1}$), $Local_{in}$ is the direct input from the landscape as it first enters the river network (M T$^{-1}$) (Supplementary Table 3), $Tributary_{in}$ is the upstream input of additional surface waters based on flow path probabilities (e.g., Supplementary Table 4, see also[41]) that enter the river along its length (M T$^{-1}$), $R$ is the removal proportion (unitless), $x$ refers to whether the input travels the entire river length (*full*) or on average half the river length (*half*), $H_L$ is the hydraulic load (L T$^{-1}$), $Q$ is the mean discharge either at the downstream end ($Q_{down}$) or the midpoint ($Q_{mid}$) of the reach length (L$^3$ T$^{-1}$), $w$ is mean river width at downstream end or midpoint (L), and $L$ is the mean length of the river order (L). We note that $v_f/H_L$ is equivalent to $k\tau$, where $k$ is the time specific uptake rate (T$^{-1}$) and $\tau$ is the residence time (T). Both forms are also equivalent to the dimensionless Damköhler number[42]. Flow path probabilities are used to identify upstream vs. tributary inputs (e.g., Supplementary Table 4). The model accounts for the removal of constituents by upstream river orders before calculating downstream concentration and uptake. Flow path probabilities are then used to calculate how the biogeochemical process accumulates with increasing watershed size.

**Scenarios.** We implemented two sets of model scenarios. The first set explored how the slope of local areal function (*m.local* in Eq. (2)) for zero-order processes affects the cumulative allometric scaling function (*d* in Eq. (1)). The second set explored how first-order processes interacting with flow conditions affects allometric scaling. All scenarios were implemented with network structure parameters from three different river network shapes derived from optimal channel networks across a range of river hydraulic parameters (network geomorphology: $R_a$, $R_b$, $R_l$; river hydraulics: $e$, $f$, $g$;[23,62] Supplementary Table 1). Parameter values are bounded by observed ranges and thus provide the basis for an envelope of potential allometric scaling functions across river network types. Across all scenarios we used constant Watershed Area = 6321 km$^2$, $A_1 = 1$ km$^2$, $L_1 = 1$ km, and mean annual runoff = 500 mm yr$^{-1}$.

Allometric scaling for cumulative function vs. watershed area (*d* in Eq. (1)) takes into account both surface area and biogeochemical activity. For the first set of scenarios, the zero-order kinetic approach was used to represent local uptake (M L$^{-2}$ T$^{-1}$) as a function of watershed area (Eq. (2)), either increasing (*m.local* > 0), decreasing (*m.local* < 0), or constant (*m.local* = 0, $y = b$). For an initial heuristic set of scenarios, *m.local* was bounded between −0.5 and 0.5 as a realistic range of possibility to explore how this parameter influences the scaling slope for cumulative function vs. watershed area. For this set of zero-order scenarios, the parameter $b$ (local rate for stream sites with watershed area = 1 km$^2$) was set to an arbitrary level (1 g m$^{-2}$ yr$^{-1}$), as it does not affect the scaling relationship when biogeochemical processes are zero-order. However, $b$ does affect the absolute magnitude of the cumulative function of surface waters.

For the second set of scenarios, first-order kinetics represented local process rates using uptake velocities (L T$^{-1}$). The 1st order kinetic scenarios were each run across a range of flow conditions, from 0.1 to 20 times mean annual flow, to explore interactions between biogeochemical demand and flow condition. For simplicity, we assumed loading concentration was constant with variation in flow (i.e., chemostatic). With the assumption of first-order reactivity, areal uptake is dependent on concentration ($U = v_f C$). Because upstream removal proportion affects downstream concentration, and removal proportions are also affected by flow conditions[3,25], with first-order kinetics the magnitude of $v_f$ is an important consideration for allometric scaling of biogeochemical function with increasing watershed area. To demonstrate scaling, we initially set $v_f = 500$ mm yr$^{-1}$ ($b$ in Eq. (2)) representative of nitrate uptake[3,38]. Although there is little empirical evidence that uptake velocities change with river size[3,38,39], in which case *m.local* = 0, some field and modeling studies do suggest lower values for larger rivers[44,69,70]. We, therefore, applied scenarios with local scaling for uptake velocity (*m.local* = −0.1, 0, and 0.1). We also explored in detail the impact of the uptake velocity constant using scenarios more typical of denitrification ($v_f = 35$ m yr$^{-1}$)[3,71] and ammonium uptake (1000 m yr$^{-1}$)[38].

Finally, we also applied an empirically derived zero-order scenario to demonstrate how allometric scaling of biogeochemical function vs. watershed size is relevant to an important issue currently being addressed by the research community, the role of surface waters in the net carbon balance in the Earth system[4,9,37]. Functions of local GPP and ER versus watershed area of the river

reach (Eq. (2)) were derived from a broad synthesis of measured stream metabolism[35] across a range of stream sizes (1–10,000 km$^2$). To the GPP and ER data in[35], we fit a linear regression to the log-log-transformed local GPP and ER vs. watershed area in pristine watersheds, resulting in areal GPP (g C m$^{-2}$ of stream bottom yr$^{-1}$) = 16.4 $A^{0.48}$ and ER (g C m$^{-2}$ of stream bottom yr$^{-1}$) = 307.1 $A^{0.22}$, where A is watershed area at the stream measurement site in km$^2$ (Supplementary Fig. 2). Recall that the constant in these equations ($b$) is equivalent to the areal process rate when $A = 1$ km$^2$, so $b$ defines processes in small headwater streams. This biological scenario was applied using the river network structure and river hydraulics in Supplementary Table 1.

**Statistical analyses**. Linear regression was applied to quantify the relationship between modeled cumulative aquatic function and upstream watershed area. Both variates were log-transformed to estimate the allometric scaling exponent ($d$ = the regression slope) for each modeled scenario. Scenarios included observed ranges in hydraulic assumptions for each river network type (i.e., rectangular, square, narrow) to characterize uncertainty in $d$ for each combination of discharge and reaction rate. Synthesis of reaction rates included median and inter-quartile range, as well as linear regression between log-transformed uptake velocity and discharge, reported in Supplementary Table 5.

## Data availability
This manuscript reports the results of a modeling study and new empirical data sets have not been generated.

## Code availability
The statistical river network model used in this analysis was developed in a Microsoft Excel ® spreadsheet with multiple scenarios run through a MATLAB wrapper. The model for the rectangular river network, the MATLAB wrapper, and a readme document are deposited on Github at the following link: https://github.com/wsag/River-Network-Statistical-Model (https://doi.org/10.5281/zenodo.5850889).

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

## Acknowledgements

This work was supported by the National Science Foundation Macrosystems Biology program: EF-1065286 (W.M.W.), EF-1065682 (W.B.B.), EF-1065055 (T.K.H.), EF-1442451 (A.M.H.), EF-1926423 (W.M.W.), EF-1926591 (W.M.W.), EF-1065255 (to PI Walter Dodds), and Long-Term Ecological Research program: PIE-LTER OCE-1637630 (W.M.W.), ARC-LTER 1637459 (W.B.B.) programs. Partial support was provided by the New Hampshire Agricultural Experiment Station (NHAES) through USDA National Institute of Food and Agriculture Hatch Project 0225006 to W.M.W. This is Scientific Contribution Number 2838.

## Author contributions

W.M.W., T.K.H., L.E.K., A.M.H., A.L.R., C.S. and W.B.B. participated in a workshop where the research was conceptualized. W.M.W., T.K.H., A.M.H. and W.B.B. secured funding. J.C.F. contributed the empirical metabolism data and analysis. A.L.R. executed the model scenarios and created Fig. 5. W.M.W. and T.K.H. wrote the initial draft of the paper. All authors contributed to writing and editing.

## Competing interests

The authors declare no competing interests.
