## [Peer Review File · Nature Communications]

Superlinear scaling of riverine biogeochemical function with watershed sizeREVIEWERS' COMMENTS

Reviewer #4 (Remarks to the Author):

I have read the R2R and the article again. My initial review of the article was in general positive, and it remains so. The authors have defended the challenges from the reviewers well in either the r2r or in the article itself. In general I think the main argument: that reactions in large rivers are disproportionately important is one that is relevant to a broad readership and that this paper presents a good mathematical argument.

A few minor comments:

Line 91. This leaves out water column processing, which perhaps could be capture with TSS surface area, or bacterial cell wall surface area, or plant surface area (after all GPP is modeled and isn't really a process that occurs strictly at the stream bottom interface).

m.local is a bit of a strange term to use over and over again (sort of the pinnacle of jargon). I would suggest trying to use something else for this term (this might also help its adoption).

Line 162. it is not clear to me what "an average value of riverine contribution to CO₂ flux (28%)" means

line 283. I wonder if you could give a sort of flow frequency for when these flow conditions are met to drive this point home a bit more.

Figure 1. Could you list the linear, superlinear and sublinear in figure A also?

Figure 3. Could you state that if these fluxes normalized to watershed area somewhere?

Reviewer #4 (Remarks to the Author):

I have read the R2R and the article again. My initial review of the article was in general positive, and it remains so. The authors have defended the challenges from the reviewers well in either the r2r or in the article itself. In general I think the main argument: that reactions in large rivers are disproportionately important is one that is relevant to a broad readership and that this paper presents a good mathematical argument.

Thank you for the positive comments on our manuscript.

A few minor comments:

Line 91. This leaves out water column processing, which perhaps could be capture with TSS surface area, or bacterial cell wall surface area, or plant surface area (after all GPP is modeled and isn't really a process that occurs strictly at the stream bottom interface).

We have now highlighted that we are not considering water column processing in this manuscript early on (L98). We also discuss how this is an additional factor towards the end of the ms that could be incorporated at a later time. We note that many of the biogeochemical processes that can be considered in this framework (gas exchange, particle settling, denitrification) do not occur within the water column while for other processes (like nutrient assimilation or GPP) they only become important in the very largest rivers.

m.local is a bit of a strange term to use over and over again (sort of the pinnacle of jargon). I would suggest trying to use something else for this term (this might also help its adoption).

We have replaced in the text m.local with the more descriptive “local scaling”, to be parallel to the “allometric scaling” used for the cumulative scaling slope, d. However, we continue to refer to m.local parenthetically when referring to scenarios using different slopes of how local areal processes change with river size.

Line 162. it is not clear to me what "an average value of riverine contribution to CO2 flux (28%)" means

This wording is changed to reflect this was an estimate of global riverine contribution to CO2 evasion from rivers.

line 283. I wonder if you could give a sort of flow frequency for when these flow conditions are met to drive this point home a bit more.

We have addressed the point regarding the detection of watershed size effects using an example, referring to different flow conditions for a particular process rate.

Figure 1. Could you list the linear, superlinear and sublinear in figure A also?

We opted not to add these labels, because the idea of superlinearity refers to whether the log-log increase is greater or less than 1, whereas the local processing slopes refer to whether the rates increases or decreases with watershed area (and are therefore greater and less than zero).

Figure 3. Could you state that if these fluxes normalized to watershed area somewhere?

The caption currently states the units are normalized per watershed area, with additional reinforcement in the main text.